Determinants of neonatal mortality in rural India, 2007–2008

Singh Aditya 1 Aditya.Singh@port.ac.uk
Kumar Abhishek 2
Kumar Amit 2
1 Global Health and Social Care Unit, School of Health Sciences and Social Work, University of Portsmouth , Portsmouth , United Kingdom
2 International Institute for Population Sciences , Mumbai , India
Erez Offer
Electronic publication date: 2013 May 28
Publication date: 2013
Volume: 1
Electronic Location ID: e75
Received 2013 Jan 16; Accepted 2013 Apr 25
Copyright: © 2013 Singh et al.
Copyright year: 2013
Copyright holder: Singh et al.
License: This is an open access article distributed under the terms of the Creative Commons Attribution License, which permits unrestricted use, distribution, and reproduction in any medium, provided the original author and source are credited.
License URL: https://creativecommons.org/licenses/by/3.0/

Keywords: Rural India, Social determinants of health, District Level Household Survey-3, Neonatal mortality

Funding: No direct financial support or funding was received to conduct this study.

==============================
Background. Despite the growing share of neonatal mortality in under-5 mortality in the recent decades in India, most studies have focused on infant and child mortality putting neonatal mortality on the back seat. The development of focused and evidence-based health interventions to reduce neonatal mortality warrants an examination of factors affecting it. Therefore, this study attempt to examine individual, household, and community level factors affecting neonatal mortality in rural India.

Data and methods. We analysed information on 171,529 singleton live births using the data from the most recent round of the District Level Household Survey conducted in 2007–08. Principal component analysis was used to create an asset index. Two-level logistic regression was performed to analyse the factors associated with neonatal deaths in rural India.

Results. The odds of neonatal death were lower for neonates born to mothers with secondary level education (O R = 0.60, p = 0.01) compared to those born to illiterate mothers. A progressive reduction in the odds occurred as the level of fathers’ education increased. The odds of neonatal death were lower for infants born to unemployed mothers (O R = 0.89, p = 0.00) compared to those who worked as agricultural worker/farmer/laborer. The odds decreased if neonates belonged to Scheduled Tribes (O R = 0.72, p = 0.00) or ‘Others’ caste group (O R = 0.87, p = 0.04) and to the households with access to improved sanitation (O R = 0.87, p = 0.02), pucca house (O R = 0.87, p = 0.03) and electricity (O R = 0.84, p = 0.00). The odds were higher for male infants (O R = 1.21, p = 0.00) and whose mother experienced delivery complications (O R = 1.20, p = 0.00). Infants whose mothers received two tetanus toxoid injections (O R = 0.65, p = 0.00) were less likely to die in the neonatal period. Children of higher birth order were less likely to die compared to first birth order.

Conclusion. Ensuring the consumption of an adequate quantity of Tetanus Toxoid (TT) injections by pregnant mothers, targeting vulnerable groups like young, first time and Scheduled Caste mothers, and improving overall household environment by increasing access to improved toilets, electricity, and pucca houses could also contribute to further reductions in neonatal mortality in rural India. Any public health interventions aimed at reducing neonatal death in rural India should consider these factors.

Introduction

The major public health interventions during the last two decades have been focused on reduction in infant and child mortality (World Health Organization (WHO), 2005; United Nations, 1995). As a result, the number of deaths among under-5 children has fallen from about 12 million to about 7.2 million during 1990–2011 (Bhutta et al., 2005; Rajaratnam et al., 2010; Lozano et al., 2011). Yet it remains a cause for concern because the annual rate of decline has been only 2.1% compared to the Millenium Development Goal-4 (MDG-4) target of 4.4% (Rajaratnam et al., 2010; Murray et al., 2007) and neonatal deaths still comprise about 40% of all under-5 deaths worldwide (Liu et al., 2012).

About 98% of all neonatal deaths occur only in developing countries while developed countries account for the rest of the neonatal deaths (Åhman & Zupan, 2007; Oestergaard et al., 2011). It has been noted that the reduction in neonatal mortality is slower than the reduction in post-neonatal and childhood mortality, particularly in low and middle-income countries (Rajaratnam et al., 2010; Black et al., 2010; United Nations International Children’s Emergency Fund, 2008; You et al., 2010; Lawn, Cousens & Zupan, 2005) which has resulted in an increase in the share of neonatal mortality in overall under-5 mortality from 39% in 1970 to 41% in 2010 (Rajaratnam et al., 2010; Lawn, Cousens & Zupan, 2005; Zupan & Aahman, 2005; United Nations (UN), 2001). Similarly, in India too, which accounts for about one-fourth of all neonatal deaths occurring around the world and has achieved substantial reductions in mortality (Office of Registrar General of India (ORGI), 2008), the share of neonatal deaths in under-five deaths has been increasing over time – from 45% in 1990 to 54% in 2010 (Rajaratnam et al., 2010). This trend indicates a slower reduction in neonatal mortality compared to post-neonatal and childhood mortality during the last two decades in India (Arokiasamy & Gautam, 2008). Nevertheless, the child survival programs in India have been focusing more on the causes of mortality and morbidity which mostly affect children in the post-neonatal period – such as pneumonia, malaria, diarrhea, and vaccine-preventable diseases (Bhargave, 2004) rather than factors such as prematurity, low birth weight and neonatal infections (Baqui et al., 2006; The Million Death Study Collaborators, 2010; Bang et al., 2005; Tinker et al., 2005).

It is argued that neonatal mortality could be reduced up to 70% only by evidence-based interventions and strategies (Darmstadt et al., 2005; Yinger & Ransom, 2003; Titaley et al., 2008). However, to adopt a focused and evidence-based approach to reduce neonatal mortality in India, a clear understanding of the associated factors is necessary. A review of past studies on this issues reveals that although there are many studies examining the factors affecting neonatal mortality available elsewhere in the world (Bhutta et al., 2005; Titaley et al., 2008; Shakya & McMurray, 2001; Samms-Vaughan, McCaw-Binns & Foster-Williams, 1990; Machado & Hill, 2003; Mahmood, 2004; Rahman & Abidin, 2010; Diallo et al., 2010), the issue seems to be understudied in India. Although there is a large body of literature available describing levels, trends and differentials in infant and child mortality at national and sub-national level (Jain, 1985; Gupta, 1990; Simmons et al., 1982; Narayana, 2008; Subramanian et al., 2006; Pradhan & Arokiasamy, 2010; Joe, Mishra & Navaneetham, 2010; Behl, 2012; Bhattacharya & Chikwama, 2011; Pandey et al., 1998), existing studies on neonatal mortality are generally limited to small geographical areas (Arokiasamy & Gautam, 2008; Kumar et al., 2013; Bapat et al., 2012; Singh, Yadav & Singh, 2012). This study, therefore, aims to examine the effect of various determinants – socio-demographic, economic, healthcare, and community – on neonatal mortality in rural India. The focus is on rural India because about two-thirds (69%) of the Indian population still lives in the rural areas (Office of the Registrar General and Census Commissioner, India, 2011) and suffers from poor early life health conditions such as high infant and child mortality compared to its urban counterpart (Ministry of Finance, 2011; Lau, Johnson & Kamalanabhan, 2012; International Institute for Population Sciences (IIPS) & Macro International, 2007).

This study is different from previous studies in three important ways. Firstly, we examine the effects of variables representing different components of the health system and socio-economic development of villages. Secondly, we try to estimate the effects of household environment separately by including variables related to toilet, water, house type and electricity in the analysis. Finally, unlike previous studies on neonatal mortality in India, we use the two-level binary logistic regression which takes into account the hierarchical structure of the data and provides correct standard errors.

Data and Methods

Ethics statement

This study uses anonymised survey data made available for academic use, for which ethical approval is not required. The survey data used in this study can be obtained by making a formal request on the official website (http://www.rchiips.org/) of the International Institute for Population Sciences, Mumbai (India) (International Institute for Population Sciences (IIPS), 2010).

Data

We use data from the third round of the District Level Household Survey (DLHS-3) conducted during 2007–08. It is a large scale, nationally representative, multi-round survey covering more than 700,000 households from 601 districts in 34 States and Union Territories of India. DLHS-3, like its former versions DLHS-1 and DLHS-2, was basically designed to provide reliable information on reproductive and child health (RCH) indicators at district level (International Institute for Population Sciences (IIPS), 2010).

The survey adopted a multi-stage stratified probability proportional to size (PPS) sampling design. The details of the survey design, implementation and response rate are given in the DLHS-3 report. The rural sample of DLHS-3 covered 559,663 households and 504,272 evermarried women of the age group 15–49 (International Institute for Population Sciences (IIPS), 2010). In this study, we use information on 171,529 infants nested in 22,587 Primary Sampling Unit (PSUs). We refer to PSU as “village” or “community” hereafter in the text.

Conceptual framework

Mosley & Chen (1984) proposed a framework that corrected the flaws in previous frameworks used by social scientists and medical scientists to study child mortality. It proposes a set of proximate determinants that directly influences the risk of child mortality. It also proposes that all other socio-economic factors must operate through this set of proximate determinants (Mosley & Chen, 1984). This framework given below has been modified for the present study (Titaley et al., 2008) and displays pathways and selected potential predictors relevant to the present study (see Fig. 1).

Figure 1 Conceptual framework showing factors affecting neonatal mortality.

Exposure variable

The neonatal death is the outcome variable in the study. It is defined as “any death occurred during first 28 completed days of life”. Neonatal death is recoded as a binary variable in this study where ‘0’ indicates that the child survived for more than 28 days and ‘1’ indicates otherwise, i.e., death of the child within 28 days. We have considered only singleton live births (i.e., all births excluding still births and twin births) in the analysis.

Independent variables

Table 1 lists all explanatory variables, their definitions and categories used in this study. These variables can be divided into four categories – community characteristics, individual/household characteristics, household environment characteristics and proximate determinants. The individual level socioeconomic variables included in this study are maternal and paternal education, maternal religion and caste, employment status of the mother and asset index.

Table 1 Operational definition and categorization of variables used in the study.

Variables	Description	
Community variables		
Accessibility by an all-weather road	Whether the village is accessible by an all-weather road – No (0), Yes (1)	
Distance to the nearest private health facility	Distance to any private health facilities (private hospital or private clinic) to the village – Within 1 km (0) = within village or within 1 km; 1–5 km (1); More than 5 km (2).	
Distance to the nearest public health facility	Distance to any public health facilities (CHC or PHC or Block PHC or PHC or Government hospital) to the village – Within 1 km (0) = within village or within 1 km; 1–5 km (1); More than 5 km (2).	
ANM/ASHA available in the village	ANM (Auxiliary Nurse and Midwife)/ASHA (Accredited Social Health Worker) resides in or visits the village – No (0), Yes (1).	
Janani Suraksha Yojana (JSY) implemented	Whether JSY has been implemented in the village – No (0), Yes (1).	
Proportion of mothers with ‘above secondary’ education	The proportion of mothers with ‘above secondary’ education in the village.	
Proportion of rich households	The proportion of rich households in the villages. It is constructed by combining two upper quintiles of the Household Wealth Index already available in the dataset.	
Region	A region in this study is a group of Indian states. North region (1) includes Jammu & Kashmir, Himachal Pradesh, Punjab, Rajasthan, Haryana, Chandigarh (Union Territory - UT) and Delhi; Central region (2) includes the states of Uttar Pradesh, Uttaranchal, Madhya Pradesh and Chhattisgarh; East region (3) includes the states of Bihar, Jharkhand, West Bengal and Orissa; North-East region (4) includes the states of Sikkim, Assam, Meghalaya, Manipur, Mizoram, Nagaland, Tripura, and Arunachal Pradesh; West region (5) includes the states of Gujarat, Maharashtra, Goa and UTs of Dadara & Nagar Haveli and Daman & Diu; South region (6) includes the states of Kerala, Karnataka, Andhra Pradesh, Tamil Nadu and the UTs of Andaman & Nicobar Islands, Pondicherry and Lakshadweep)	
Socioeconomic variables		
Mother’s education	Mother’s education is defined based on years of schooling and divided into four categories – Illiterate (0) = 0 years of schooling; Primary (1) = 1–5 years of schooling; Secondary (2) = 6–10 years of schooling; Above secondary (3) = more than 10 years of schooling.	
Father’s education	Father’s education is defined based on years of schooling and divided into four categories – Illiterate (0) = 0 years of schooling, Primary (1) = 1–5 years of schooling; Secondary (2) = 6–10 years of schooling; Above secondary (3) = more than 10 years of schooling.	
Asset index	The asset index is constructed using principal component analysis and divided into three categories – Poor (0); Middle (1); Rich (2).	
Religion	Religion is divided into three categories – Hindu (0); Muslim (1); Others (3) = all religious groups other than Hindu and Muslim.	
Caste/Tribe	Caste/Tribe is divided into four categories – Scheduled Castes – SC (0); Scheduled Tribes – ST (1); Other Backward Castes – OBC (2); General (3).	
Employment of the mother	Mother is said to be employed if a mother was engaged in any economic activity in last 12 months preceding survey. It has been divided into three categories – Agriculture worker, farmer, and labourer (0); Unemployed (1); Professional/service/production workers (2).	
Improved source of water	Whether the household has access to piped water within the premises of the house – No (0); Yes (1).	
Improved toilet facility	Whether the household has access to improved toilet facility – No (0); Yes (1).	
House type	Type of house – Kaccha (0) = wall, floors, and roofs are kaccha; Pucca (1) = walls, floors, and roofs are pucca.	
Electricity	Whether the household has an electricity connection – No (0); Yes (1).	
Proximate Determinants		
Mother’s age at birth	Maternal age at birth is divided into four categories – <20 years (0); 20–24 years (1); 25–29 years (2); ≥ 30 years (3).	
Sex of the child	Sex of the child – Girl (0); Boy (1).	
Tetanus toxoid (TT) Injection	Number of TT injection taken during pregnancy – One (0); Zero (1); Two or more (2).	
Iron and folic acid (IFA) tablets/syrup	Consumption of adequate IFA tablets/syrup during pregnancy – No (0); Yes (1).	
Birth order	The order in which the child was born – First order (0); Second order (1); Third order (2); Four and above (3).	
Delivery complications	Whether mother faced any complication/s during delivery – No (0); Yes (1).	
Place of delivery and skilled birth attendance (SBA)	A variable is computed with combination of place of delivery and assistance during delivery by any health personnel. This is divided into three categories – Home (0); Home + SBA (1) = home delivery assisted by any trained health personnel; Health facility (2).	
Antenatal Care (ANC) visits	This variable is computed using two variables, the frequency and the timing of ANC visits. This is divided into four categories – No visit (0) = no ANC visit; First trimester + ≥ 4 visits (1) = Four or more visits in first trimester; First trimester + < 4 visits (2) = less than four visits in first trimester; ANC visits made in second and third trimester (3).	
Notes.

Categories in italics have been used as reference category in the regression models.

The asset index in this study has been used as a proxy for economic status of the household (Montgomery et al., 2000; Filmer & Ptitchett, 2001; Vyas & Kumaranayake, 2006; O’Donnell et al., 2008; Rutstein, 2008; Howe, Hargreaves & Gabrysch, 2009). The asset index is based on variables related to household amenities. The variables included are – mattress, cooker, chair, sofa set, cot, table, fan, radio, black & white television, color television, sewing machine, mobile phone, telephone, fridge, watch, bicycle, scooter, cart, car, tractor, pump, thresher, cooking fuel used, landholding, and number of rooms in the house. However, it does not include variables representing household environment (type of house, availability of electricity, access to improved water and sanitation) because we wish to separate the effects of these variables later in the analysis. We use the Principal Component Analysis (PCA) to construct the index. The index is divided into three categories – Poor, Middle, Rich.

Statistical Analysis

The contingency table and two-level logistic regression analysis are used to examine the factors affecting neonatal mortality in rural India. Descriptive analysis is used to understand the differentials in neonatal deaths across the selected covariates. Unadjusted odds of neonatal mortality are then calculated, and only statistically significant variables are retained for the subsequent analysis. To estimate the adjusted effects of different individual, household and community level factors on neonatal mortality, we use a two-level binary logistic regression (Guang & Hongxin, 2000). We choose a two-level regression technique instead of simple regression analysis because it can take into account the hierarchical structure of the study sample in which individuals (children) are nested within communities (PSUs). Multilevel models allow for such datasets and produce standard errors (SEs) adjusted for clustering of observations (Diez-Roux, 2000; Heck & Thomas, 2009; Blakely & Woodward, 2000). On the other hand, if SEs are estimated using simple binary logistic regression model, there is a chance of underestimation of SEs which could affect the interpretation of the results.

Before applying the two-level regression, we examined the extent to which the outcome of interest varies at higher levels. We fitted a null model and carried out the Wald test to know whether residuals at village level are statistically significant (results not shown). We found that the Wald statistic was highly significant (result not shown) which suggested that fitting a two-level model made sense in this context (Rasbash et al., 2009). We looked for evidence of multicollinearity using variance inflation factor (VIF) as a post-estimation procedure. It initially revealed that the variable measuring antenatal care (ANC timing and frequency) had a very high VIF. Therefore, we removed it from the regression analysis. The small value of VIF (1.68) from the final regression model indicated the absence of any significant collinearity among the variables. The result of logistic regression is presented in the form of odds ratios with statistical significance shown by p-values. The statistical analysis for this study was performed with the help of statistical software Stata 12 SE and MLwiN 2.24 (Stata Corporation, 2011; Rasbash et al., 2009).

Results

To identify the associated predictors of neonatal mortality in rural India, 171,529 singleton live births to currently married women within 3 years preceding the survey (2004–07) were included in the analysis as the study population. We found that 2892 neonatal deaths occurred, which was 1.68% of total singleton live births, during this period.

Table 2 Characteristics of variables.

Variables	Birthsb	% Birthsc	Neonatal deaths	% Neonatal deathsd	
Community characteristics	
Region	
Central	52537	29.8	1216	2.30	
North	10736	15.2	129	1.20	
East	56965	23.1	1003	1.80	
North-East	21141	12.6	229	1.10	
West	13756	9.4	134	1.00	
South	16394	9.9	218	1.30	
Accessibility by an all-weather road	
No	26208	15.1	434	1.63	
Yes	144764	84.9	2491	1.70	
Distance to the nearest private health facility	
Within 1 km	31,540	18.6	504	1.59	
1–5 km	19,128	11.1	324	1.67	
>5 km	120,204	70.3	2,097	1.72	
Distance to the nearest public health facility	
Within 1 km	31,136	18.3	484	1.54	
1–5 km	2,438	1.4	27	1.09	
>5 km	137,329	80.3	2,413	1.73	
ANM/ASHA available/visiting the village	
No	142530	83.0	2480	1.72	
Yes	28442	17.0	445	1.54	
JSY implemented in the village	
No	138795	81.18	2419	1.72	
Yes	32167	18.82	506	1.55	
Individual/socio-economic characteristics	
Mother’s education	
Illiterate	85924	49.3	1710	1.97	
Primary	26512	15.6	458	1.70	
Secondary	54422	32.4	728	1.32	
>Secondary	4663	2.8	33	0.69	
Father’s education	
Illiterate	47758	27.5	991	2.05	
Primary	29087	17.0	548	1.86	
Secondary	83091	48.7	1260	1.49	
>Secondary	11585	6.8	130	1.09	
Asset index	
Poor	60496	34.8	1253	2.05	
Middle	56268	32.9	962	1.68	
Rich	54724	32.4	714	1.29	
Religion	
Hindu	131729	76.5	2355	1.76	
Muslim	21936	12.6	376	1.71	
Others	17852	10.9	198	1.09	
Caste/Tribe	
Scheduled Castes	33069	19.5	676	2.02	
Scheduled Tribes	34631	21.2	469	1.32	
Other Backward Castes	67595	39.6	1267	1.86	
General	32937	19.7	461	1.39	
Employment of the mother a	
1	95210	55.4	1490	1.91	
2	9943	5.8	156	1.54	
3	66166	38.7	1280	1.54	
Household environment	
Improved source of water	
No	56415	33.0	804	1.40	
Yes	115106	67.0	2125	1.82	
Improved toilet facility	
No	125912	72.9	2409	1.89	
Yes	45606	27.1	520	1.13	
House type	
Kaccha	140470	81.8	2543	1.78	
Pucca	31050	18.2	386	1.23	
Electricity	
No	79810	45.3	1639	2.04	
Yes	91711	54.7	1290	1.39	
Proximate determinants	
Mother’s age at birth	
<20	21972	12.8	538	2.42	
20–24	68773	40.3	1159	1.66	
25–29	48555	28.3	674	1.36	
>30	32221	18.6	558	1.71	
Sex of the child	
Girls	79437	46.3	1223	1.52	
Boys	92070	53.7	1699	1.82	
Tetanus Toxoid injection	
1	54765	31.6	1086	1.96	
0	10750	6.5	241	2.17	
2 or more	105343	61.9	1592	1.49	
Iron and Folic Acid tablets/syrup	
No	81154	47.4	1396	1.69	
Yes	90302	52.6	1533	1.67	
Birth order of the child	
1	52975	30.1	1135	2.10	
2	44187	18.5	616	1.37	
3	29064	8.8	392	1.33	
4 and above	45250	42.6	780	1.70	
Delivery complications	
No	63862	37.8	926	1.42	
Yes	107659	62.2	2003	1.84	
Place of delivery and SBA	
Home	98407	56.7	1654	1.66	
Home + SBA	9754	5.7	146	1.47	
Health Facility	63287	37.6	1127	1.75	
Time and frequency of ANC visits	
No Visit	51491	30.1	1027	1.97	
First trimester + ≥ 4 visits	33231	20.4	424	1.26	
First trimester + < 4 visits	30683	18.1	493	1.59	
Second or third semester visits	53038	31.3	948	1.77	
Total	171456	100.0	2929	1.69	
Notes.

ANM/ASHA = Auxiliary Nurse and Midwife/Accredited Social Health Worker; SBA = Skilled Birth Attendance; ANC = Antenatal Care; JSY = Janani Suraksha Yojana (Mother Protection Scheme).

a Employment of the mother: 1 = Agricultural worker/farmer/labourer, 2 = Unemployed, 3 = Professional/service/production worker.

b Some variables had missing cases.

c The percentage of birth is calculated using total number of births i.e. the sample of this study (171456).

d The percentage of deaths is the percentage of neonatal deaths out of total number of births in the subgroup. For example – the per cent deaths for Central region (2.30) comes from dividing ‘Neonatal deaths’ (1216) with ‘Births’ (52537) in the Central region.

The characteristics of the study variables are presented in Table 2. Around 50% of neonates were born to mothers who were illiterate. Only 6% of children were born to mothers working as professionals/service/production workers. A great majority of neonates were born to mothers living in kachcha or semi-kachcha houses (81%) and without any improved sanitation facilities (71%). About 12% of children were born to adolescent mothers and about three-fourths to Hindu women. About 30% mothers never had an antenatal check-up, 38% of mothers did not have adequate IFA and about 47% did not receive a TT injection. A little more than 60% of deliveries occurred at home. About 62% of children were born to women who suffered from at least one delivery related complication.

Tables 3–6 present crude and adjusted odds ratios for neonatal mortality according to background characteristics. Unadjusted odds ratios revealed that some variables like accessibility by an all-weather road, place of delivery and consumption of adequate IFA did not turn out to be statistically significant. We dropped these variables in further analysis. The results of two-level logistic regression revealed that there was a great variation in the odds of neonatal mortality by region. Lower odds of neonatal death were observed in almost all the regions compared to the Central region. The odds of neonatal death were 19% lower in rich villages than in poor villages. It was also seen that the odds of neonatal death in villages, where the nearest government health facility is located one to five kilometers away, were 33% lower than the villages where the nearest public health facility is located within one kilometer from the village (Table 3).

Table 3 Odds ratios for neonatal death according to community characteristics and region of residence.

Independent variables	Unadjusted	Adjusted©	
	OR	CI (95%)	p	OR	CI (95%)	p	
		Lower	Upper			Lower	Upper		
Region	
Central®	1.00				1.00				
North	0.71	0.63	0.79	<0.001	0.89	0.78	1.00	0.056	
North-East	0.72	0.66	0.80	<0.001	0.64	0.53	0.77	<0.001	
East	0.46	0.40	0.53	<0.001	0.69	0.62	0.76	<0.001	
West	0.42	0.35	0.50	<0.001	0.51	0.42	0.62	<0.001	
South	0.57	0.49	0.65	<0.001	0.73	0.62	0.86	<0.001	
Community characteristics	
Accessibility by an all-weather road	
No®	1.00								
Yes	1.04	0.94	1.15	0.457					
Distance to the nearest public health facility	
Within 1 km®	1.00				1.00				
1 to 5 km	1.06	0.92	1.22	0.409	0.97	0.83	1.13	0.719	
More than 5 kms	1.09	0.99	1.21	0.074	0.97	0.87	1.08	0.592	
Distance to the nearest private health facility	
Within 1 km®	1.00				1.00				
1 to 5 km	0.71	0.48	1.05	0.084	0.67	0.45	1.01	0.056	
More than 5 kms	1.13	1.03	1.25	0.013	0.97	0.87	1.09	0.617	
ANM/ASHA available in the village	
No®	1.00				1.00				
Yes	0.90	0.81	0.99	0.037	1.05	0.94	1.18	0.353	
Janani Suraksha Yojana implemented	
No®	1.00				1.00				
Yes	0.90	0.82	0.99	0.034	1.06	0.96	1.17	0.268	
Proportion of mothers with above secondary education	0.43	0.37	0.49	<0.001	0.87	0.70	1.09	0.229	
Proportion of rich households	0.49	0.43	0.55	<0.001	0.81	0.65	1.00	0.049	
Notes.

p = p-value; CI (95%) = Confidence Interval at 95% level; OR = Odds Ratio; ANM/ASHA = Auxiliary Nurse and Midwife/Accredited Social Health Worker. © The logistic regression model controlled for the following variables as well – mother’s education, father’s education, asset index, religion, caste/tribe, employment status of the mother, improved source of water, improved toilet facility, house type, electricity, mother’s age at birth, sex of the child, birth order of the child, timing and number of ANC visits, Tetanus Toxoid injection, Iron and Folic Acid tablets/syrup, delivery place and SBA, and delivery complications.

Table 4 Odds ratios for neonatal death according to socioeconomic characteristics.

Independent variables	Unadjusted	Adjusted©	
	OR	CI (95%)	p	OR	CI (95%)	p	
		Lower	Upper			Lower	Upper		
Socioeconomic characteristics	
Mother’s education	
Illiterate©	1.00				1.00				
Primary	0.87	0.78	0.96	0.007	1.01	0.90	1.14	0.823	
Secondary	0.67	0.61	0.73	<0.001	0.96	0.85	1.09	0.563	
>Secondary	0.35	0.25	0.50	<0.001	0.60	0.41	0.88	0.009	
Father’s education	
Illiterate©	1.00				1.00				
Primary	0.91	0.82	1.01	0.067	0.98	0.88	1.10	0.75	
Secondary	0.73	0.67	0.79	<0.001	0.85	0.76	0.94	0.00	
>Secondary	0.54	0.45	0.64	<0.001	0.76	0.61	0.94	0.01	
Asset index	
Poor©	1.00				1.00				
Middle	0.82	0.76	0.90	<0.001	1.00	0.91	1.10	0.967	
Rich	0.63	0.57	0.69	<0.001	1.04	0.90	1.19	0.612	
Religion	
Hindu©	1.00				1.00				
Muslim	0.96	0.86	1.07	0.444	0.95	0.83	1.08	0.397	
Others	0.62	0.53	0.71	<0.001	0.99	0.83	1.17	0.881	
Caste/Tribe	
Scheduled Castes©	1.00				1.00				
Scheduled Tribes	0.66	0.58	0.74	<0.001	0.72	0.63	0.82	<0.001	
Other Backward Castes	0.92	0.83	1.01	0.066	0.94	0.85	1.04	0.207	
General	0.68	0.60	0.77	<0.001	0.87	0.77	0.99	0.041	
Employment status of the mothera	
1©	1.00				1.00				
2	0.81	0.87	0.75	0.012	0.90	0.83	0.98	0.012	
3	0.81	0.95	0.68	<0.001	1.00	0.84	1.19	0.991	
Notes.

p = p-value; CI (95%) = Confidence Interval at 95% level; OR = Odds Ratio; © The logistic regression model controlled for following variables as well – region, accessibility by an all-weather road, distance to the nearest private health facility, distance to the nearest public health facility, ANM/ASHA available in the village, Janani Suraksha Yojana implemented, proportion of mothers with above secondary education, proportion of rich households, improved source of water, improved toilet facility, house type, electricity, mother’s age at birth, sex of the child, birth order of the child, timing and number of ANC visits, Tetanus Toxoid injection, Iron and Folic Acid tablets/syrup, delivery place and SBA, and delivery complications.

a Employment status of the mother: 1 = Agricultural worker/farmer/labourer, 2 = Unemployed, 3 = Professional/service/production.

Table 5 Odds ratios for neonatal death according to household environmental factors.

Independent variables	Unadjusted	Adjusted©	
	OR	CI (95%)	p	OR	CI (95%)	p	
		Lower	Upper			Lower	Upper		
Household environment	
Improved source of water	
No	1.00				1.00				
Yes	1.30	1.20	1.41	<0.001	1.13	1.04	1.24	0.006	
Improved toilet facility	
No	1.00				1.00				
Yes	0.59	0.54	0.65	<0.001	0.87	0.77	0.98	0.019	
House type	
Kachcha	1.00				1.00				
Pucca	0.68	0.61	0.76	<0.001	0.87	0.77	0.98	0.025	
Electricity	
No	1.00				1.00				
Yes	0.68	0.63	0.73	<0.001	0.84	0.76	0.92	<0.001	
Notes.

p = p-value; CI (95%) = Confidence Interval at 95% level; OR = Odds Ratio. © The logistic regression model controlled for the following variables as well – region, accessibility by an all-weather road, distance to the nearest private health facility, distance to the nearest public health facility, ANM/ASHA available in the village, Janani Suraksha Yojana implemented, proportion of mothers with above secondary education, proportion of rich households, mother’s education, father’s education, asset index, religion, caste/tribe, employment status of the mother, mother’s age at birth, sex of the child, birth order of the child, timing and number of ANC visits, Tetanus Toxoid injection, Iron and Folic Acid tablets/syrup, delivery place and SBA, and delivery complications.

Table 6 Odds ratios for neonatal death according to proximate determinants.

Independent variables	Unadjusted	Adjusted©	
	OR	CI (95%)	p	OR	CI (95%)	p	
		Lower	Upper			Lower	Upper		
Proximate determinants	
Mother’s age at birth	
Below 20	1.00				1.00				
20–24	0.68	0.62	0.76	<0.001	0.85	0.76	0.95	<0.001	
25–29	0.56	0.50	0.63	<0.001	0.74	0.65	0.85	<0.001	
30 and above	0.70	0.62	0.79	<0.001	0.87	0.74	1.01	0.065	
Sex of the child	
Girl	1.00				1.00				
Boy	1.20	1.12	1.29	<0.001	1.21	1.12	1.30	<0.001	
Birth order of the child	
1	1.00				1.00				
2	0.65	0.58	0.71	<0.001	0.65	0.59	0.72	<0.001	
3	0.62	0.56	0.70	<0.001	0.58	0.51	0.66	<0.001	
4 and above	0.80	0.73	0.88	<0.001	0.62	0.55	0.71	<0.001	
Timing and number of ANC visits	
No ANC visits	1.00								
First trimester + > = 4 ANC visits	0.64	0.57	0.71	<0.001					
First trimester + < 4 ANC visits	0.80	0.72	0.89	<0.001					
2nd or 3rd semester ANC visits	0.89	0.82	0.98	0.014					
Tetanus Toxoid injection	
One	1.00				1.00				
No	0.88	1.02	0.77	0.083	0.75	0.65	0.87	<0.001	
Two or more	0.67	0.77	0.58	<0.001	0.65	0.56	0.75	<0.001	
Iron and Folic Acid tablets/syrup	
No	1.00								
Yes	0.99	0.92	1.06	0.719					
Delivery place and SBA	
Home	1.00								
Home but SBA	0.88	0.75	1.05	0.176					
Health facility	1.06	0.98	1.14	0.131					
Delivery complications	
No	1.00				1.00				
Yes	1.29	1.19	1.39	<0.001	1.20	1.10	1.30	<0.001	
Notes.

p = p-value; CI (95%) = Confidence Interval at 95% level; OR = Odds Ratio; SBA = Skilled Birth Attendance; ANC = Antenatal Care. © The logistic regression model controlled for the following variables as well – region, accessibility by an all-weather road, distance to the nearest private health facility, distance to the nearest public health facility, ANM/ASHA available in the village, Janani Suraksha Yojana implemented, proportion of mothers with above secondary education, proportion of rich households, mother’s education, father’s education, asset index, religion, caste/tribe, employment status of the mother, improved source of water, improved toilet facility, house type, electricity.

At the individual level, the mother’s education was significantly associated with a reduction in the odds of neonatal deaths. However, this is not true for all literate or educated women. There appears to be a threshold number of years of schooling needed for a significant reduction in neonatal mortality. Infants born to mothers with more than 10 years of schooling were about 40% less likely to experience neonatal death compared to those born to illiterate mothers. The same is true about the father’s education. The odds of neonatal mortality reduced significantly by 15% and 24% among children whose fathers had their schooling up to ‘secondary’ and ‘above secondary’ level, respectively, compared to children belonging to illiterate fathers. The caste of the child also emerged as a significant predictor. The odds of neonatal death reduced by 28% and 13% in Scheduled Tribes and Others category of castes, respectively, compared to Scheduled Caste children. The odds decreased significantly by 10% among the neonates of unemployed mothers compared to the neonates of those working as farmers/laborers/agricultural workers (Table 4).

All household environment variables appeared as significant predictors of neonatal mortality even after controlling for other factors. Children from households with access to an improved source of water were 13% more prone to death in the neonatal period than those belonging to households with no accessibility to an improved source of water. Having an improved toilet facility and electricity in the household reduced the odds of neonatal death significantly by 13% and 16% as compared to the household where these facilities were not available. The odds of neonatal death decreased by 13% among children belonging to households living in a pucca house compared with those living in kachcha houses (Table 5).

Among the proximate determinants, all variables included in the analysis were found to be significant except the variables for time and frequency of ANC visits of the mother although results were in the expected direction. Increasing mother’s age at birth reduced the odds of neonatal death. The odds decreased significantly by 15% and 26% respectively among children whose mothers were 20–24 and 25–29 years old, respectively, at the time of their birth than children of adolescent mothers. Boy neonates in rural India were found to be 21% more prone to neonatal death compared to girl neonates. Another demographic variable found significantly related to the reduced risk of neonatal mortality was their birth order. The odds of neonatal death reduced by 35%, 42%, and 38% for second, third and ‘four and above’ birth orders, respectively, compared to the first birth order. In comparison to the mothers who received only one TT injection during pregnancy, the odds of neonatal death were significantly lower (OR = 0.65, p = 0.00) among those infants whose mothers had two or more TT injections. Similarly, mothers who did not receive any TT injections, their infants were too about 25% less at the risk of death in the neonatal period compared to those who had received one TT injection. The risk of neonatal death increased by 20% if the mother experienced any delivery complications compared to those mothers who did not experience any of the delivery complications (Table 6).

Discussion

In this study, we used the most recent data available in the public domain to examine the factors affecting neonatal mortality in rural areas of India. Estimates based on two-level logistic regression model indicate that a number of factors were significantly associated with neonatal mortality. Our findings revealed that maternal education significantly reduced the odds of neonatal death in rural India. The finding is similar to previous studies that have established a link between a mother’s education and the child’s survival (Basu & Stephenson, 2005; Caldwell, 1979; Mellington & Cameron, 1999; Ware, 1984; Zanini et al., 2011). Maternal education is argued to improve child health through increased knowledge about the practices to improve child health (Caldwell, 1979) and increased use of maternal care services (Elo, 1992; Raghupathy, 1996). Similarly, the father’s education was also found important for reduction in neonatal deaths.

Results indicated that neonates belonging to STs and ‘Others’ caste groups were less likely to die before one month compared to SC children. STs have remained one of the most socioeconomically deprived communities in India for centuries (Borooah, 2005). A large majority of them live in inaccessible and far-off places which are still underdeveloped (Mohindra & Labonté, 2010). Yet, significantly lower odds of deaths compared to ST neonates appears quite strange and is a matter of further investigation. The lower risk of neonatal death among ‘Others’ neonates compared to SC neonates is not surprising because ‘Others’ castes have been economically better off and socially and politically privileged (Zacharias & Vakulabharanam, 2011).

Children belonging to mothers who stayed at home (unemployed) were less likely to die during the neonatal period compared to the children belonging to mothers who worked as farmers/agricultural workers/laborers. The finding is similar to that of previous studies (Titaley et al., 2008; Kishor & Parasuraman, 1998). It is worth mentioning here that unemployed mothers in rural India were more educated (44% versus 19%) and richer (40% versus 18%) than those who worked as farmers/agricultural workers/laborers (data not shown in tables). This coupled with enough available time for seeking antenatal care and taking care of her neonate (like breastfeeding) could explain the significant decline in the odds of neonatal death (Basu & Basu, 1991; Hobcraft, 1993; Tulasidhar, 1993). On the other hand, there was no significant difference between the odds of neonatal mortality among mothers who worked as professional/service/production workers and farmers/agricultural workers/laborers. This is supported by the findings of previous studies (Zanini et al., 2011; Murthi, Anne-Catherine & Jean, 1995).

All four variables – improved source of drinking water, improved sanitation, type of house, and availability of electricity – included to represent the household environment appeared as significant predictors of neonatal deaths in rural India. Access to improved water actually increased the risk of neonatal death in rural India. It is worth noting here that the relationship of access to an improved source of drinking water with neonatal mortality has been ambiguous. It has shown both positive (Mahmood, 2004) and negative effects (Zanini et al., 2011) on neonatal mortality. At first, it seems to be a peculiar result in itself. Newborn babies after all are not directly affected by the source of water. Nevertheless, it is plausible that they are indirectly affected. In the case of rural India, the access to improved water sources like a hand-pump within the premises probably leads to more use of water compared to the households where the source of water is located away from the house. However, in the absence of proper drainage, (only 4% of Indian households had any underground or covered pucca drainage system in 2011) the household wastewater stagnates or stays in the open drainages in and around the house (NSSO 58th round) (Office of the Registrar General and Census Commissioner of India, 2011). This coupled with mud floors (according to Census of India 2011, about 62% rural houses have mud floors) create an infectious environment which could help spread malaria, diarrhea, and other infectious diseases in both the mother and the newborn (Rath et al., 2010). In previous studies too, the two waterborne diseases – maternal malaria among pregnant mothers (causes anemia in mothers during pregnancy and subsequent low birth weight of the newborn) and diarrhea among neonates – have been found to be among the main causes of neonatal death in the developing countries (Hartman, Rogerson & Fischer, 2010; Yilgwan, Hyacinth & Oguche, 2011; Rijken et al., 2012; Yakoob et al., 2011; Lawn et al., 2009; Ghosh, 2010; Titaley et al., 2010; Taha, Gray & Abdelwahab, 1993). Since, the purpose of this study is not to catalogue and investigate the different channels through which the source of water could affect the chances of neonatal death, further exploration is needed on this issue.

Three other variables representing household environment – availability of improved toilets, pucca house and electricity – were found to reduce the likelihood of neonatal death. Access to improved toilet reduces the risk of dying through the mechanism of less exposure of neonates to contamination making them less susceptible to diseases and infections, and eventually death (Rahman & Abidin, 2010). Unlike kachcha houses, pucca houses have hardened brick walls and concrete/brick roofs which provide better shelter from harsh weather conditions especially during monsoon season. Availability of electricity may help to create better environmental conditions in the house for the newborn (Poel & Doorsleaer, 2009). It not only helps in hygienic preparation of food but also encourages the use of electric fan, television and radio.

Among five proximate determinants included in the analysis, the mother’s age was found to be significantly associated with reduction in neonatal mortality (Titaley et al., 2008; Zanini et al., 2011). Older mothers not only possess better knowledge of pregnancy and childbirth but also enjoy greater autonomy compared to younger mothers which help them take care of their neonates in a better way in this period (O’Malley & Forrest, 2002; Hobcraft & Kiernan, 2001). It also emerges from the analysis that the risk of neonatal death decreases with increasing birth order of the child. These results confirm the results of many studies of the past conducted in different settings around the world (Titaley et al., 2008; Zanini et al., 2011; Rutstein, 1984; Yerushalmy, 1938; Hussain, 2002; Rahman & Huq, 2009). A strong association has been previously reported between the sex of the child and neonatal mortality (Rahman & Abidin, 2010; Rahman & Huq, 2009; Chaman et al., 2009; Machado & Hill, 2005). Similarly, in this study too, we find that the boys are more susceptible to death within the first month after birth compared to girls. It has been argued that boys are biologically weaker than girls due to various reasons (Ulizzi & Zonta, 2002; Green, 1992; Mahy, 2003). These reasons include immunodeficiency (Green, 1992) leaving baby boys more vulnerable to infectious diseases (Arokiasamy & Gautam, 2008), late maturity (Alonso, Fuster & Luna, 2006) resulting in a higher prevalence of respiratory diseases in males, and congenital malformations of the urogenital system.

The main causes of neonatal mortality are intrinsically linked to the health of the mother and the care she receives during pregnancy and delivery. Our findings indicate that one of the components of antenatal care (TT injection) is significantly associated with lower risk of neonatal deaths. Our study confirmed the results of previous studies that using two or more TT injections during pregnancy help reducing neonatal deaths substantially through reducing the likelihood of tetanus infection in newborns (Taha, Gray & Abdelwahab, 1993; Rahman et al., 1982; Gupta & Keyl, 1998; Yusuf et al., 1991; Blencowe et al., 2010; Arnold, Soewarso & Karyadi, 1986). It has been noted that neonatal tetanus is one of the major causes of neonatal deaths in developing countries (Taha, Gray & Abdelwahab, 1993; Rahman et al., 1982; Gupta & Keyl, 1998; Vandelaer et al., 2003). Being an effective strategy to reduce the number of maternal and newborn deaths due to tetanus, increasing the coverage of TT injections could be an important intervention in rural India.

It is well established now that delivery complications cause poor neonatal outcomes as indicated by low apgar scores and low arterial cord blood pH. Confirming the same, our study also found that the neonates born to women, who experienced complications like vaginal bleeding, fever or convulsions during delivery, had remarkably higher odds of neonatal death compared to those born to women without any complications during delivery. However, higher odds of neonatal deaths can also be attributed to the mothers’ inability to take care of their newborn properly in the postnatal period as they take time to recover from damage due to complications during birth. The findings are consistent with many other studies in the South Asian setting (Titaley et al., 2008; Titaley, Dibley & Roberts, 2011; Mercer et al., 2006).

At the community level, the prosperity of the villages (as measured by the proportion of richest households in the PSU) had a significant influence on neonatal mortality. It is generally argued that community factors, such as overall level of wealth and education in the community, may influence the individual’s behaviour, partly through social learning and social influence. It has also been argued that if mothers in a community are more wealthy, they are likely to be more educated and have better knowledge of health care behaviour. Their knowledge and attitudes may be passed on to other women. It is very much possible in a rural Indian setting, where communities are socially more cohesive than urban India. The consequences of such social influence and learning from educated mothers may include better nutrition, adequate and timely vaccination, home care, a hygienic household environment, and interaction with health workers (Kravdal, 2004; Parashar, 2005; Moursund & Kravdal, 2003).

Quite surprisingly, we found that any increase in the distance to the nearest private health facility decreased the odds of neonatal death. Though it is inconsistent with most of the previous studies conducted in different settings around the world, a study in Pakistan, a neighboring country, has found similar patterns (Noorali, Luby & Rahbar, 1999). Such results might be attributed to purposeful outreach by health workers or some other unknown situations, however, the issue needs a further exploration.

The ‘region’ of residence is also significantly associated with the risk of neonatal death. It was found that neonates from ‘South’ and ‘West’ regions were less likely to die in the neonatal period. Higher levels of socioeconomic development and better functioning of the healthcare system could be some of the factors behind the better performance of states in these regions. The states covered under the Central regions included Madhya Pradesh and Uttar Pradesh (including Uttarakhand, Chhattisgarh). These states are characterized by comparatively poor socioeconomic and demographic indicators and dysfunctional government healthcare systems. Hence, it is not surprising that most of the regions show lower odds of neonatal death compared to the Central region.

Limitations of the study

Although this study identified important determinants of neonatal mortality in rural India, it has a few limitations. Firstly, we could not include many other community level variables that possibly have an effect on neonatal mortality because they were not available in the dataset that we used. Such variables might include service supply environment such as quality, quantity, and the adequacy of the services; beliefs and traditions about pregnancy and motherhood prevailing in the community. Secondly, some variables like employment of the mother and asset index represented the conditions of the time of the interview, not of the time when the child was born.

Conclusion

To conclude the study, we can say that the growing share of neonatal mortality in under-5 mortality warrants adoption of comprehensive strategies to further reduce the neonatal mortality in rural India. Although a continuum of healthcare during pregnancy, childbirth, and even during the postnatal period (Arokiasamy & Gautam, 2008; Titaley et al., 2008) is necessary for further reductions in neonatal mortality, ensuring uptake of an adequate quantity of TT injections during pregnancy should be a priority in maternal and child health related programmatic interventions and strategies (Singh et al., 2012; Ayaz & Saleem, 2010). Certain groups of children and women, such as neonates of first birth order, neonates belonging to Scheduled Castes, adolescent mothers and mothers working in agricultural sector need special attention. Targeting these groups in order to provide the continuum of essential maternal and childcare would be a crucial step if neonatal mortality in rural India will be further reduced. In addition to that, improving the overall household environment by increasing access to improved toilets, electricity and pucca houses could also contribute to further reductions in neonatal mortality in rural India.

Supplemental Information

Supplemental Information 1 Raw data

Click here for additional data file.

Additional Information and Declarations

Competing Interests

Author Contributions

The authors have declared that no competing interests exist.

Aditya Singh conceived and designed the experiments, performed the experiments, analyzed the data, contributed reagents/materials/analysis tools, wrote and edited the paper.

Abhishek Kumar conceived and designed the experiments, performed the experiments, analyzed the data, contributed reagents/materials/analysis tools, wrote the paper.

Amit Kumar analyzed the data and wrote the paper.

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
