# Peer review of "Determinants of neonatal mortality in rural India, 2007–2008"

_PeerJ, doi:10.7717/peerj.75_

## Round 0.1 · original submission · Minor Revisions

The study presented herein is important in highlighting the need for a better collaborative structure of the perinatal followup of pregnant women. Please modify your manuscript according to the reviewers suggestion

·

Basic reporting

This article is a population based study, addressing an important issue concerning neonatal death in rural India, using a large and detailed database, adding to the importance of the results. Generally, the article is too long, especially in the introduction and methods section. The introduction should only briefly address the worldwide knowledge and importance of the subject, and focuses mainly on relevant data concerning neonatal and child death in India. As for the methods section, the author provided too many details that seem to be of local importance, and are unwarranted in this setting. For that matter, table 1 is too detailed.

In the result section care should be taken for describing only details involving general public interest and not local interest (for example: comparison between central and other regions should not include a detailed list of all states, already provided in table 1.

Table 2, which holds the major results of this study, is unclearly labeled. The percentage of birth is from the total number of births, whereas the percentage of deaths is the percent of neonatal death from the number of births in the relevant subgroup, and not from total number of births. Since both parameters are aligned together, it is worthwhile to clarify this in the table or in its’ comments. In addition p value in this table, and in the result section in general, may be better presented as smaller than a certain value (p<0.001 etc.) and not as equaling zero (p=0.00).

In the discussion, some of the major results are presented adequately, with reference to other relevant data from previous studies, whereas other topics are awkwardly discussed, giving the impression that the collaboration with other medical professionals from the disciplines of neonatology and OBGYN might have been beneficial. For example, the discussion regarding maternal vaginal bleeding, fever or convulsions during pregnancy and a higher rate of neonatal deaths, relating it to the fact that the mother’s inability to take care of the newborn, without discussing the well-established poor neonatal outcome (such as apgar scores, cord arterial blood pH) resulting from maternal obstetrical complications such as placental abruption, chorioamnionitis and eclampsia, presenting as the above clinical signs.

Experimental design

No comments

Validity of the findings

The large database on which this study is based gives this article power and validity. This article concerns the important issue of neonatal death in rural areas in India, and it’s conclusions could be adopted to similar rural areas around the world.

Additional comments

In addition, there are several typing, spelling and grammatical errors that mandate additional proofreading, for example:
• In the last sentence of the first paragraph of the introduction the ‘4.4%’ has a typing error
• Figure 1 uses abbreviations not addressed earlier in the text (such as TT, ANC) and therefore should be clarified in the legend
• In table 2 – the word ‘birth’ is misspelled in the heading of the table
• In table 1 – under ‘mothers age at birth’ – a missing ‘a’ in the word ‘at’

·

Basic reporting

no comment

Experimental design

no comment

Validity of the findings

no comment

Additional comments

This is a very interesting study presenting Indian neonatal mortality related factors and opening a door for intervention.

---

## Round 0.2 · Minor Revisions

The manuscript has improved substantially and I have minor comments to the authors:
1- The Introduction section is still too long, please try to shorten it to about 1-1.5 A4 pages.
2- Tables - Please add to Table 3 the variables the Odds ratio were adjusted for. I would like to suggest to split this table into 2-3 smaller tables that will increase uits readability

---

## Round 0.3 · accepted · Accept

The authors have addressed the minor comments and the paper is ready for publication